# Novel Magnetic Inorganic Composites: Synthesis and Characterization

**DOI:** 10.3390/polym13081284

**Published:** 2021-04-15

**Authors:** Marco Natali, Sergio Tamburini, Roberta Bertani, Daniele Desideri, Mirto Mozzon, Daniele Pavarin, Federico Spizzo, Lucia Del Bianco, Federico Zorzi, Paolo Sgarbossa

**Affiliations:** 1ICMATE, CNR, Corso Stati Uniti 4, 35127 Padova, Italy; marcostefano.natali@cnr.it (M.N.); sergio.tamburini@cnr.it (S.T.); 2Department of Industrial Engineering, University of Padova, Via Marzolo 9 (P.S., R.B., M.M.), Via Gradenigo 6/a (D.D.) and Via Venezia 1 (D.P.), 35131 Padova, Italy; roberta.bertani@unipd.it (R.B.); daniele.desideri@unipd.it (D.D.); mirto.mozzon@unipd.it (M.M.); daniele.pavarin@unipd.it (D.P.); 3Department of Physics and Earth Science, Polo Scientifico Tecnologico, University of Ferrara, Via G. Saragat 1, 44122 Ferrara, Italy; federico.spizzo@unife.it (F.S.); lucia.delbianco@unife.it (L.D.B.); 4CEASC, Via G. Jappelli 1/A, 35121 Padova, Italy; federico.zorzi@unipd.it

**Keywords:** magnetic geopolymer composites, SrFe_12_O_19_ particles, morphological characterization, magnetic properties, impedance measurements

## Abstract

The addition of magnetic particles to inorganic matrices can produce new composites exhibiting intriguing properties for practical applications. It has been previously reported that the addition of magnetite to concrete improves its mechanical properties and durability in terms of water and chloride ions absorption. Here we describe the preparation of novel magnetic geopolymers based on two different matrices (G1 without inert aggregates and G2 with inert quartz aggregates) containing commercial SrFe_12_O_19_ particles with two weight concentrations, 6% and 11%. The composites’ characterization, including chemical, structural, morphological, and mechanical determinations together with magnetic and electrical measurements, was carried out. The magnetic study revealed that, on average, the SrFe_12_O_19_ magnetic particles can be relatively well dispersed in the inorganic matrix. A substantial increase in the composite samples’ remanent magnetization was obtained by embedding in the geopolymer SrFe_12_O_19_ anisotropic particles at a high concentration under the action of an external magnetic field during the solidification process. The new composites exhibit good mechanical properties (as compressive strength), higher than those reported for high weight concretes bearing a similar content of magnetite. The impedance measurements indicate that the electrical resistance is mainly controlled by the matrix’s chemical composition and can be used to evaluate the geopolymerization degree.

## 1. Introduction

The addition of magnetic particles to inorganic matrices has been explored to prepare new composites exhibiting intriguing properties for practical applications (see Magment^®^ [1]). The addition of magnetite (Fe_3_O_4_) to concrete has been proved to improve its mechanical properties (i.e., split tensile, flexural, and compressive strength [2]) and durability in terms of water and chloride ions absorption [2].

Concretes bearing different amounts of Fe_3_O_4_ or Fe_3_O_4_@SiO_2_ nanoparticles also behave as electromagnetic wave absorbers (particularly of microwaves) and show γ-rays shielding properties, thus featuring a huge potential application in the construction of electromagnetic wave interference buildings [3,4,5,6]. The replacement of natural coarse aggregates with magnetite aggregates added in large amount to cement, fly ash, ground granulated blast furnace slag, and silica fume produce heavyweight self-compacting concretes and their addition under alkaline activation has been used to obtain heavyweight geopolymer concretes that provide proper protection from sources emitting harmful radiation in medical and nuclear industries [7,8,9]. An intriguing advantage derives from the synthetic versatility of concretes and geopolymers in terms of the possible introduction of a wide variety of additives able to absorb or dissipate electromagnetic waves [10,11,12].

Geopolymers [13] are inorganic binders different from regular concretes since they do not form calcium silicate hydrates for strength but are based on the polycondensation of silica and alumina with high alkali content. They have attracted the attention of researchers from different engineering fields owing to their modular environmentally friendly preparation procedure (low temperature, no harmful gas emission) [14], feasibility to incorporate high volumes of wastes or metal particles to produce value-added building materials with excellent physical and mechanical properties [15,16,17]. As an example, the incorporation of fly ash into geopolymers has been demonstrated to improve thermal stability and mechanical properties, thus providing an experimental basis for the utilization of solid waste soda residues and recycling of waste concrete [18,19]. As for the mechanical properties, it is known that they are improved by the incorporation and the homogeneous distribution of nanomaterials due to the formation of a denser microstructure, but the identification of the optimum content of nanomaterials is very important to achieve the respective high performance in strength as well as workability [20]. In the case of the addition of nanosized Fe_2_O_3_, it was reported that in Portland cement composites, a homogeneous microstructure was obtained with magnetite loading lower than 10% *w/w* with an improvement of the compressive strength by 60%. Higher added amounts produced aggregation of the nanoparticles with a detrimental effect on the mechanical performance of the material [9].

Recently, geopolymers containing magnetite or maghemite nanoparticles have been prepared and applied in environmental remediation processes, such as water decolorization [21,22], arsenic [23] or heavy metals [24] removal from aqueous solutions.

Geopolymers and magnetic geopolymers have also been proposed as low-cost, effective catalysts for environment governance by “end-of-pipe” treatments, such as photocatalytic degradation of dye wastewater or gas purification and have been employed for hydrogen and biodiesel production as well as in green heterogeneous catalytic processes [25,26].

The preparation of magnetic inorganic composites has also attracted considerable interest as candidate materials for high-frequency electronic components, such as miniaturized antennas and inductors [27,28].

It is to note that recently a concrete based on SrFe_12_O_19_ particles dispersed in alumina cement has been investigated as a sacrificial material for ex-vessel core catcher in nuclear power plants [29,30], and SrFe_12_O_19_/MFe_2_O_4_ (M = Zn, Ni) composites have been shown to exhibit microwave absorption properties [4,31].

In the present work, the preparation of novel magnetic geopolymers, based on two different matrices containing commercial SrFe_12_O_19_ particles, with two different concentrations, is described. The objective was to investigate the influence of composition and experimental conditions for preparation on the morphology, dispersion of SrFe_12_O_19_ particles, electrical and mechanical properties. Particularly attractive is the study of the magnetic properties of the composites and of their modification when solidification of the geopolymers is performed in the presence of an external magnetic field. The research project’s target is to design magnetic inorganic composites to apply for pollutant removal [32,33,34] and heterogeneous catalytic processes [35] on a large scale.

## 2. Materials and Methods

### 2.1. Materials

Reagents used for geopolymer preparation were: (a) metakaolin (MK) (medium particle size 1.2 μm, BET 6.031 m^2^/g) produced in-house by calcining in an oven for 5 h commercial kaolinite powder (MiMac Srl, CE, Italy) at 750 °C; (b) granulated blast furnace slag (GBFS) from the ILVA Metallurgical Plant (Taranto, Italy) reduced to a grain size of 60–400 μm before use; (c) potassium silicate activator (Ksil) with a molar modulus (MM) SiO₂/K₂O of 1/1.3 and dry matter concentration of 45%, prepared in-house by mixing a 50% w colloidal silica dispersion and KOH pellets with distilled water at least 24 h before use.

The commercial magnetic particles of SrFe_12_O_19_ ferrite (Industrie ILPEA s.p.a.) were micrometric in size and polycrystalline. Two different types of Sr ferrite particles were used, namely isotropic particles (product code: P02-Sr-D), is-SrFe_12_O_19_, and anisotropic particles (product code: P11), anis-SrFe_12_O_19_.

The SrFe_12_O_19_ ferrite has a hexagonal crystalline structure and is a hard ferrimagnetic phase with magnetocrystalline anisotropy coefficient K = 3.6·10^6^ erg/cm^3^ [36], and an easy axis parallels to the c crystallographic axis [37]. In each of the isotropic particles, the crystallite orientation is random, and therefore, no preferential magnetocrystalline anisotropy direction exists. This implies the existence of a magnetic configuration characterized by magnetic domains with different orientations (for simplicity of description, it can be assumed that the magnetic domains coincide with the crystallites). On the contrary, each anisotropic particle is textured; namely, the crystallites have a preferred orientation resulting in the appearance of a preferential anisotropy direction, i.e., of a magnetization easy axis. In this case, a magnetic configuration with parallel domains is predictable.

### 2.2. Preparation of the Samples

New inorganic magnetic composites were prepared by using an alkaline activation process carried out in the presence of commercial magnetic particles. Details of sample preparation are as follows: MK, slag, k-silicate (SiO_2_/K_2_O = 1.3, dry matter = 45%) and H_2_O were mixed by hand in a beaker, using the proportions indicated in Table 1, using a spatula; this process was assisted by pneumatic vibration for 3 min to start the reaction of MK and slag reagents with the activator solution.

At this point, inert aggregates and magnetic particles could be added; then, the mixture was further mixed under vibration for about 10 min, and a dense, uniform and highly thixotropic paste was formed. The paste was dispensed with a spatula into the appropriate sample molds for each sample type and further vibrated briefly to homogenize the paste inside the mold. Samples were hardened by curing at 60 °C in an oven for 24 h keeping the molds inside a sealed plastic bag at saturated humidity provided by a wet cloth inside the bag. After curing and demolding, samples were further stored for 7 days in a sealed plastic bag.

Two different matrices were prepared: a matrix labeled G1, where no inert aggregates were added, and a matrix labeled G2, including inert quartz aggregates in an amount of 53% of the total dry weight of the blend. Quartz aggregates were added to the matrix to try to reduce the risk of aggregation of magnetic particles. Reagents and mix ratios of the two matrices are reported in Table 1. Magnetic particles were added with a mass concentration of about 6% and about 11% to both G1 and G2 matrices by keeping approximately constant the amount of extra water at about 10% in mass for G1 and at about 6–7% for G2. For matrix G1, the 6% composition yielded a very fluid mixture as the amounts of aggregates were very low. Therefore, in this case, additional samples labeled as DENSE were prepared by reducing the amount of added water to 2–3% in mass, thus keeping a workability (thixotropy) mixture comparable to that of the 11% composition.

The labeling of the geopolymer samples containing isotropic and anisotropic SrFe_12_O_19_ ferrite magnetic particles includes “is” and “anis”, respectively (Scheme 1).

As described in more detail in Section 2.3.2, some of the magnetic geopolymer samples were prepared by applying an external magnetic field during the curing process to study whether and to what extent this affected the spatial and magnetic configuration of the embedded particles. Hence, the name of magnetized samples included the “M” suffix (Scheme 1). The list and labeling of the samples are reported in Table 2.

### 2.3. Characterization Techniques

#### 2.3.1. Chemical, Structural, and Morphological Characterization

The microanalysis and the samples’ morphology were studied by environmental scanning electron microscopy (ESEM) with an embedded Quanta 200 FEI-EDX embedded.

The density of samples was measured according to UNI EN 206-2016 [38].

Fourier-transform infrared spectroscopy (FTIR) spectra of powders were obtained on a Nicolet spectrophotometer Avatar 320 as KBr pellets. For each sample, 32 scans were recorded in the range 4000–400 cm^−1^ in the transmittance mode with a resolution of 4 cm^−1^.

Raman spectroscopic determinations were carried out using a continuous wave linearly polarized (514.5 nm) wavelength, 2.41 eV, 16 mW power, instrument. The laser beam was focused by a 100 × objective lens, resulting in a spot of about 1 μm in diameter.

Solid-state NMR experiments, carried out on the non-magnetic materials, were performed on a Bruker AVANCE III spectrometer 300 (magnetic field of 7.0 T corresponding to ^27^Al and ^29^Si Larmor frequencies of 78.066 and 59.623 MHz, respectively) equipped for solid-state analysis in 4 mm diameter zirconia rotors with Kel-F caps. The magic angle was accurately adjusted before data acquisition using KBr. ^29^Si chemical shifts were externally referenced to solid tetrakis(trimethylsilyl)silane at −9.8 ppm (in relation to TMS), and ^27^Al chemical shifts are externally referenced to AlCl_3_·6H_2_O (0 ppm). The semiquantitative 29Si single-pulse experiments were collected at a spinning frequency of 6 kHz, a recycling delay of 60 s and 2000 transients. ^27^Al experiments were collected at a spinning frequency of 13 kHz with a pulse of 1.0 µs and a recycle time of 2 s. About 2000 scans were needed using a single pulse experiment. The signal patterns of the spectra were deconvoluted with the DMFT program using the Gaussian/Lorentz curve, which leads to the best result [39].

X-ray diffraction (XRD) measurements were carried out using a PANalytical X’Pert Pro diffractometer equipped with a Co X-ray tube and a real-time multiple strip (RTMS) detector (X’Celerator). Panalytical High Score Plus software and Panalytical ICSD database were used for phase identification. Full-scale XRD patterns are given in Appendix A. Meanwhile, in the paper are reported XRD patterns with reduced angular ranges containing only the main peaks of the phases of interest.

Thermogravimetric measurements with a differential thermal analyzer (TG-DTA) were carried out on a Netzsch STA 449 thermoanalytical equipment in the 30–1200 °C range in alumina crucibles in air and with a heating rate of 10 K/min, using neutral alumina as reference material.

#### 2.3.2. Magnetic Characterization

For the magnetic study, for each type of sample, two cylindrical specimens were prepared with dimensions of 5 mm (diameter) × 30 mm (length) inside small polyethylene sealed tubular containers. One of the cylinders, for some selected samples, was solidified under magnetic field by placing it, while in the fluid state, between two cubic (2 × 2 × 2 cm^3^) SmCo_5_ block magnets kept apart at 3.5 cm of distance and keeping the cylinder between the magnets during the subsequent curing stage at 60 °C for about 12 h. The magnetic field H_APP_ experienced by the sample was ~3.5 kOe. Magnetic hysteresis measurements were performed using a Quantum Design superconducting quantum interference device (SQUID) magnetometer at temperatures T = 20 K and 300 K, with a maximum applied field of 50 kOe. Both the isotropic and anisotropic commercial SrFe_12_O_19_ magnetic particles were investigated in the form of powder. As for the magnetic geopolymers, a piece of 5 mm in length, suitable for SQUID analysis, was cut at the center of each of the prepared cylinders.

#### 2.3.3. Electrical Characterization

The geopolymer electrical characterization was performed by doing measurements of impedance on samples prepared in the following way. Disk samples of diameter 5 cm and thickness around 1 cm were prepared with formulations: G1_matrix, G1is_6_norm, and G1is_6_highW, as well as a sample G1is_6_highK using a higher amount of k-silicate, Ksil (weight rations MK:slag: Ksil = 1:1:2 instead of 1:1:1.33) (Table 3).

The configuration of a parallel plate capacitor with a dielectric between its plates was realized. Starting from the geopolymer disk samples, the two opposite circular surfaces of each disk were painted with silver conductive paint and then a rectangular piece of copper tape, with conductive glue and a wire welded on it, was put on each one of the two opposite painted surfaces. The equivalent electrical model of this configuration was a resistor and a capacitor in parallel (R-C parallel), plus an inductor, connected in series with the R-C parallel, for the electrical model of the two wires. Measurements were performed by using two LCR meters, a GW Instek LCR-6300 (frequency range 10 Hz–300 kHz) and an Agilent 4285A (frequency range 75 kHz–30 MHz). The two instruments have shown a very good agreement in the frequency range 75 kHz–300 kHz, where both instruments could be used. The inductor that modeled the electrical behavior of the two connection wires (each one about 8 cm long) was roughly estimated as the inductance of 100 nH, value based on the assumption of an inductance per-unit-length of 1 µH/m. Impedance experimental data, in the frequency range 10 Hz–30 MHz, have shown that the inductive contribution above indicated could be neglected and therefore, the final model was given by only the R-C parallel. The measured resistance and capacitance were used for the estimation of the conductivity and relative permittivity of each sample. With the geometry and by using the relation of the resistance of a cylindrically shaped uniform material, the estimation of the conductivity was obtained. With the geometry and the relation of the capacitance of a parallel plate capacitor, the estimation of the relative permittivity was calculated. The electrical conductivity of the ferrite magnetic powders in the explored frequency range (10 Hz–30 MHz) was not expected to significantly contribute to conductivity because of its very low conductivity [40], several orders of magnitude lower than our geopolymer matrix, and the small percentage of added powder.

#### 2.3.4. Mechanical Characterization

For mechanical measurements, a series of rectangular prism samples of dimensions 22 × 22 × 44 mm^3^ were prepared to determine their compressive strength. Mechanical measurements of compressive strength were performed in triplicate on samples aged for 7 days after curing; in this case, the samples were not magnetized during the molding process. Compressive strength was measured using an Unitronic S205 test machine (Matest, BG, Italy) equipped with a 50 kN load cell.

## 3. Results and Discussion

### 3.1. Characterization of the SrFe_12_O_19_ Particles

It is known that the structural and magnetic properties of SrFe_12_O_19_ particles may change depending on the specific synthesis route [41,42,43]. Therefore, we have carried out a characterization of the commercial Sr hexaferrite particles used to prepare the composite geopolymer samples.

The morphology of the Sr ferrite particles was investigated by ESEM. The results are shown in Figure 1.

The anisotropic particles are in the form of platelets that are often stacked (Figure 1a,b), with a thickness of some tenths of micron along the c crystallographic axis, as indicated by the producer, and a lateral size of some microns. The isotropic particles are in the form of platelets too (Figure 1c,d), but, compared to the anisotropic ones, they appear slightly smaller on the average and show a somewhat more rounded shape, possibly due to a lower crystallinity degree, i.e., a smaller size of the crystalline grains. The Energy Dispersive X-ray analyses (EDX) showed that isotropic and anisotropic SrFe_12_O_19_ powders had a different average composition, as reported in Table 4, with a higher Ba content in the anisotropic powder. A detailed analysis of single particles of the anisotropic powder showed that they contain variable amounts of Ba. The ESEM images showed that, in the case of the isotropic powder, some groups of larger particles are present where the Ba content is lower than the detection threshold.

In both the FTIR spectra of is-SrFe_12_O_19_ and anis-SrFe_12_O_19_ (Figure 2), the characteristic peaks at around 620 and 540 cm^−1^ (the asterisks in Figure 2) together with those at around 600 and 440 cm^−1^ (the circles in Figure 2, assigned to the Fe-O bending of Fe-O_4_, tetrahedral, and Fe-O stretching of Fe-O_6_, octahedral, respectively) are present. The slight difference in the relative intensities of the peaks can be related to the different shapes and compositions [44]. There is no evidence of adsorbed water in both cases.

In Figure 3, the XRD patterns of is-SrFe_12_O_19_ and anis-SrFe_12_O_19_ are reported. Both profiles show typical peaks of hexagonal SrFe_12_O_19_ [45], which match with the reference pattern 98-018 4961 (Panalytical ICSD database) in the case of isotropic powder and are also accompanied by the peaks of the Ba_0.98_Fe_11.93_O_18.84_ (reference pattern 98-008-7406) in the case of the anisotropic ones. The presence of Ba ferrite in anis-SrFe_12_O_19_ is in agreement with the results of the EDX analysis (Table 4). The common impurities of unreacted Fe_2_O_3_ and SrCO_3_ precursors are present in very low amounts.

The magnetic hysteresis loops of both the is-SrFe_12_O_19_ and the anis-SrFe_12_O_19_ particles are shown in Figure 4. Nominal values of the saturation magnetization for the SrFe_12_O_19_ phase are M_S_ ~74 emu/g at room temperature and M_S_ ~104 emu/g at 0 K [37].

As reported in Table 5, in the anisotropic particles, we have measured M_S_ ~75 emu/g and M_S_ ~107 emu/g at T = 300 K and T = 20 K, respectively, in agreement with the expected values. In fact, the presence of traces of Ba ferrite, revealed by XRD (Figure 3), does not affect the magnetization since Sr and Ba ferrites have similar magnetic properties [46]. The values of M_S_ for the isotropic particles are slightly smaller, namely ~69 emu/g at 300 K and ~97 emu/g at 20 K.

The loops exhibit a wasp-waist-like shape, i.e., they narrow at the center, the effect being particularly evident in the isotropic particles. This is often observed in samples made of hard and soft magnetic elements mixed together [47,48]. In our case, this feature may indicate that the Sr ferrite powders are magnetically not homogeneous and consist of particles with different anisotropy, possibly due to some stoichiometric and structural disorder, as revealed by EDX analysis (Table 4). This would also account for the reduced M_S_ value measured in is-SrFe_12_O_19_ [49,50]. However, the wasp-waist-like shape may appear more or less pronounced depending not just on the relative fractions of different magnetic particles but also depending on their respective magnetization and coercivity and on how they are mixed and possibly interact through both exchanges and dipolar interactions. It is quite well demonstrated that the magnetic moments of fully exchange-coupled soft and hard nanocrystallites reverse coherently under an applied magnetic field, giving rise to a loop typical of a single magnetic phase, whereas a constricted loop can be observed in the case of incomplete coupling [51,52,53].

It has recently been shown that colloids of mixed iron oxide nanoparticles with different coercivity values, although of the same order of magnitude (ratio ~1:3), exhibited magnetic loops that did not reveal the presence of two distinct populations, actually [54]. Wasp-waist-like loops were measured in assemblies of nanoparticles, consisting of the same magnetic phase, due to the action of dipolar interactions only [55,56]. Hence, the casuistry is very wide and articulated. In the perspective of comparing the loops in Figure 4 for the starting is-SrFe_12_O_19_ and anis-SrFe_12_O_19_ particles and those measured in the geopolymer samples shown below (Section 3.3.3), we can generally say that the observation of a wasp-waist-like hysteresis loop in a certain assembly of magnetic particles is a sort of fingerprint of a specific arrangement and/or magnetic interacting state of the particles themselves.

From the loops, we have derived the squareness parameter, indicated with M_R_, corresponding to the ratio between the remanent and the saturation magnetization values. M_R_ is higher in the anisotropic particles, both at 300 K and 20 K (Table 5).

According to the Stoner and Wohlfarth model for a ferromagnetic element with uniaxial anisotropy, a perfectly squared hysteresis loop (i.e., M_R_ = 1) is measured by applying the magnetic field parallel to the anisotropy axis [57]. The same occurs for an assembly of these magnetic elements, provided that they are not interacting and that their anisotropy axes are all parallel to the applied magnetic field. However, in the case of an assembly of magnetic elements noninteracting and with random spatial orientation of the anisotropy axes, M_R_ = 0.5.

The high values of M_R_ found in the anisotropic Sr-ferrite particles indicate that their preferential anisotropy axes, i.e., the c crystallographic axes, were quite well aligned with the external field during the loop measurement. Indeed, this is quite unexpected since we did nothing to obtain this degree of spatial order of the particles. It should be considered that to be analyzed by SQUID, a small number of SrFe_12_O_19_ was put at the bottom of a polymer capsule and slightly pressed to avoid that they moved during the measurement. The capsule had an elongated shape, and during the measurement, the field was applied parallel to its major axis. Hence, a likely explanation is that, due to their flattened shape, the particles, forming stacks as observed by ESEM, lie on the bottom of the capsule. The isotropic platelets are expected to show the same tendency. However, a lower M_R_ was measured in this case because the isotropic particles do not possess a preferential anisotropy axis. M_R_ is slightly smaller than 0.5, in line with the Stoner and Wohlfarth model, since the whole set of crystallites, of which the particles consist, represents an assembly of magnetic elements with randomly oriented anisotropy axes.

This description may also account for the lower H_C_ measured in is-SrFe_12_O_19_ compared to that of anis-SrFe_12_O_19_ (Table 5) since the Stoner Wohlfarth model predicts that the coercivity of the random assembly is about half of that measured in the ordered one. In particular, at T = 300 K, we measured H_C_ ~0.95 kOe in is-SrFe_12_O_19_ and H_C_ ~2.11 kOe in anis-SrFe_12_O_19_. However, in the case of our samples, the situation is more complex because the particles are very close to each other and so certainly subjected to magnetic interactions, which are known to alter the coercivity compared to that of an assembly of isolated particles. In particular, dipolar interactions tend to decrease H_C_ [55,56,58], and the same can be observed in exchange-coupled particles [59,60]. Moreover, the is-SrFe_12_O_19_ and anis-SrFe_12_O_19_ particles exhibit constricted hysteresis loops (Figure 4). Therefore, the H_C_ values of the isotropic and anisotropic particles should not be compared in the framework of the Stoner and Wohlfarth description.

### 3.2. Characterization of the G1 and G2 Matrices

In Figure 5, the ESEM images of the G1 and G2 matrices are shown. The composition reported in Table 6, as an average of at least five measurements, agrees with the presence in matrix G2 of added quartz; in both cases, the morphology appears highly porous, with G1 less compact than G2.

The FTIR spectra of the matrices (Figure 6) show the presence of relatively broad signals centered at 1040, 860, and 450 cm^−1^ (the asterisks in Figure 6), characteristic of the aluminosilicate structure [61]. In the G2 matrix, the sharp absorptions at 795, 777 and 690 cm^−1^ of quartz (the circles in Figure 6) are also visible [62].

The Raman spectra (Figure 7) show for both matrices the weak broad signals at 530 cm^−1^ of the Q2 Si-O-Si system and at 650 cm^−1^ of the calcium silicate hydrate, together with a weak, narrowband at 797 cm^−1^ due to SiO_4_^4−^ groups (the asterisks in Figure 7). Absorptions at 1085 cm^−1^ due to the O-C-O symmetric stretching of CaCO_3_ precipitate as calcite are accompanied in the case of G2 by strong absorptions at 464, 205, and 127 cm^−1^ (the circles in Figure 7) due to quartz [63].

MAS-NMR spectroscopic studies conducted on the ^27^Al and ^29^Si nuclei have been essential to defining the type of geopolymeric matrix obtained [64,65].

^27^Al MAS-NMR of the well-shaped signal of the geopolymers G1 and G2 at around +58.2 ppm has been associated with tetrahedral aluminum [66] with the almost complete transformation of the kaolinitic and gehlenite slag phases in the Ca-based geopolymer matrix [67,68], suggesting three-dimensional networking of the type AlQ4(4Si) (Figure 8).

As for the ^29^Si MAS-NMR spectra of G1 and G2, the reaction of MK and GBFS with alkaline silicate solution produced a broad peak centered at −87.4 ppm (Figure 9) for both matrices that are indicative of the new geopolymer matrix formed.

In the presence of metakaolin, an aluminosilicate network dominates the reaction creating cyclic structures that were analyzed deconvolving the spectra to show the components forming the broad peak. In G1, the tall peak at −91.26 ppm could be attributed to the new Q4(3Al) and at −86.6 ppm to the Q4(4Al) aluminosilicate network. In G2, the Gaussian symmetric peaks at −87.3/−88.0 ppm show an undetermined presence of both aluminosilicate networks. At −71.6 and −74.5 ppm for the two matrices and at −106.8 ppm for quartz in G2, the unreacted GBFS contributions are visible.

In both the XRD patterns of the matrices (Figure 10), a very broad hump between 20 and 40°, characteristic of amorphous materials, is visible. The XRD diffractograms show the same mineralogical phases, but G1 contains only small traces of vaterite, and G2 contains much more than G1 [13].

The TG-DTA measurements of the geopolymer matrices (Figure 11) reveal a total water content of 15.1% and 7.5%, respectively, and an asymmetric mass-loss peak in the 50–230 °C temperature range. A broad endothermic peak in the DTA curve centered at about 132 °C (G1) and 112 °C (G2), extending for both to about 225 °C, corresponds to a weight loss of 8.5% for G1 and 4.2% for G2 associated with free or “interstitial” water loss [69,70]. In the range, 225–510 °C, a total water loss of 3.8% and 1.7% for G1 and G2, respectively, continuous and featureless, was associated with bound water and dihydroxylation of hydroxyl groups. In the range, 510–715 °C, the amount of hydroxyl lost (2.6% and 1.5%) is more similar in the two matrices than that lost in the range 30–510 °C, indicating that the higher quantity of water used in the synthesis of G1 respect to G2 remained mainly interstitial. Above 725 °C, no additional mass loss occurs with no sintering/melting events to 1200 °C. The endothermic peak of G2 DTA at 574 °C is in agreement with phase changing of quartz, while the broad exothermic peak at about 1000 °C indicates in both samples that the residual unreacted metakaolin crystallized into mullite [71].

### 3.3. Structural and Morphological Study of the Magnetic Inorganic Composites

#### 3.3.1. Morphology and Thermal Behavior

The density of G1 (2.02 g/cm^3^) was lower than G2 (2.29 g/cm^3^) due to the different amounts of water added and increased in the presence of the magnetic particles.

In Figure 12, some selected ESEM images of the magnetic composites show the porosity of the materials, with apparently no influence of the geopolymerization carried out under magnetic conditions. The image in Figure 13a of G2anisM_6, detected using backscattered electrons (thus, the embedded magnetic particles appear to be lighter concerning the surrounding matrix), shows that the particles have the tendency to form aggregates. It is to note that in the case of G2anisM_6, the magnetic particles give rise to stacks (Figure 13a), otherwise in the G2anis_6 sample, particles tend to clump together (Figure 13b).

Magnetic particles also aggregate in the G2isM_6 sample (Figure 13c), but in a lower extent, giving rise to smaller aggregates (typically 20 × 20 μm for is-SrFe_12_O_19_ vs. 30 × 30 μm for anis-SrFe_12_O_19_, in the presence of the same amount of ferrite, in more distributed shape) and appear to be more distributed in the case of G2is_6, where rare, ordered aggregates can be detected (Figure 13d).

As observed in Figure 11, the TG-DTA curves of the two matrices concerning the composites G1iso_11 and G2iso_11 show an overlapping trend.

#### 3.3.2. FTIR, Raman and XRD Characterization of the Magnetic Geopolymers

In the FTIR of the composites, the signals of the SrFe_12_O_19_ at 552 and 601 cm^−1^ are evident, in particular, in the G1 derivatives, while the absorption at 650 and 540 cm^−1^ are masked by the signals of the matrix (Figure 6).

The Raman spectra of the composites do not differ from the corresponding matrices. A sharpening of the peaks in the samples containing anis-SrFe_12_O_19_ polymerized under magnetic field and the presence of a peak at 688 cm^−1^, which is the most intense peak of the ferrite [44], can be observed (Figure 7).

In Figure 14, the XRD diffractograms of G1is_11 and G1isM_11, in the 2θ range from 35° up to 44° are reported together with those of G2anis_6 and G2anisM_6, where the two main peaks of ferrites are labeled with the corresponding measured d-spacings. The peaks of SrFe_12_O_19_ and Ba0.98Fe11.93O18.84 coincide with the d-spacings of their relative reference patterns 98-018 4961 and 98-008-7406 (Panalytical ICSD database) even after the geopolymerization process and the peaks broadening (FWHM) shows no significant variation. This means that the process affects neither the ferrites’ structures nor their crystallite sizes and/or microstrain (which is related to the peak broadening).

#### 3.3.3. Magnetic Characterization

The saturation magnetization M_S_ was measured at T = 300 K in all the prepared geopolymer samples. By calculating the ratio between the M_S_ values of the sample and of the starting Sr ferrite particles, the mass concentration of the magnetic particles was evaluated. We found that, in all the samples, the concentration of magnetic particles was in agreement with the nominal one reported in Table 2, within the error. As this result turned out to be independent of the specific matrix, in Table 7, the data corresponding to four representative samples are reported.

On these samples, we also have measured M_S_ at T = 20 K (Table 7). The ratio between the M_S_ measured at 300 K and at 20 K is consistent, within the error, with that calculated for the starting magnetic particles. Hence, the incorporation of the particles in the geopolymer matrix did not affect the magnetization and its thermal dependence, which means that there was no interdiffusion between the magnetic and geopolymeric phases.

In Figure 15a, the magnetic hysteresis loops at T = 300 K of the four selected samples are shown; they are normalized to their M_S_ values to better compare their shapes. The loops are similar and appear almost superposed. The wasp-waist-like shape, characterizing the loops of the starting magnetic particles (Figure 4), is no more visible. This indicates that a different magnetic arrangement of the particles and possibly a different state of magnetic interaction was obtained concerning the starting ones, namely that the existence of large agglomerates that substantially reproduce the behavior of the non-dispersed starting particles can be excluded. The M_R_ values are very close in all the samples, slightly smaller than 0.5 (Table 7), which reveals that a random spatial orientation of the magnetic particles in the geopolymer matrix, both isotropic and anisotropic, was attained during the synthetic process. Moreover, H_C_ is higher in the composite samples compared to the starting particles, which is consistent with a reduction of the strength of the magnetic interactions between the particles after being dispersed in the geopolymer matrix, namely with an increased inter-distance. Hence, the analysis of the magnetic loops indicates that a good dispersion of the particles in the matrix was obtained.

Magnetic hysteresis loops have also been measured at T = 300 K on geopolymer samples with compositional properties similar to those of the four selected ones addressed above but prepared in H_APP_ = 3.5 kOe. In particular, the measurements were carried out with the magnetic field along the same direction of H_APP_. The results are shown in Figure 15b. The loops are more squared than those in Figure 15a relative to the samples prepared in H_APP_ = 0. In fact, M_R_ is larger than 0.5 in all the samples (Table 8).

This is explained considering that H_APP_, applied when the Sr ferrite particles were dispersed in the liquid matrix before solidification, activated two different magnetizing mechanisms: it magnetized, mainly through the movement of the domain walls, the particles to a certain extent so that each acquired a net magnetic moment; it exerted a torque acting on the magnetic moments resulting in the physical rotation of the particles themselves [72]. Therefore, a high degree of alignment of the magnetic moments of the Sr-ferrite particles, both anisotropic and isotropic, was obtained. When the geopolymeric matrix solidified, and H_APP_ was removed, the remanent moments of the particles maintained their preferential alignment, parallel to H_APP_.

This has resulted in higher M_R_ values compared to those measured in the samples solidified without H_APP_. In particular, the highest M_R_ = 0.84 was measured in G1anisM_11 (Figure 16), while M_R_ = 0.6 in the other samples (experimental error 2%). Similarly, the H_C_ parameter (Table 8) is almost the same in all the samples (H_C_ ~2.4 kOe), except in G1anisM_11, where it reaches the highest value of 4.10 kOe. Both results indicate that a better alignment of the remanent moments was achieved in the case of anisotropic particles dispersed in the matrix at high concentration, whereas substantially similar configurations were attained in the other samples. To account for this effect, it is to be considered that the particles, when magnetized by H_APP_, produced an internal magnetic field (i.e., Lorentz field [46]), which added to H_APP_, parallel to it. The anisotropic particles, larger and with higher M_S_ than the isotropic ones, acquired larger moments and were more easily rotated by H_APP_, thus producing a more intense internal field. This mechanism, together with the high concentration of anisotropic particles present in G1anisM_11, gave rise to an internal field strong enough to determine a substantial improvement in the alignment of the magnetic moments compared to the other investigated samples.

#### 3.3.4. Electrical Characterization

To shed light on the interaction of the composite geopolymer with electromagnetic fields, its electrical impedance was measured. The AC impedance spectra (Nyquist plots), carried out on the samples listed in Table 3, recorded about 1 year from sample preparation, are shown in Figure 17 by plotting the imaginary part of the impedance Z, Zim, versus the real one, Zre.

In the plots, the high-frequency arcs are well outlined. In all cases, the arc center is depressed below the real axis, which is due to the relaxation of polarization processes within the material and results in dielectric dispersion, i.e., a decrease of the capacitance when increasing the frequency [73]. The depression angle is in the range of 23–27° for the four plots reported.

From the plots, the values of the “bulk resistance” corresponding to the real part of impedance at where the high and low-frequency arcs met and of the “relaxation frequency” corresponding to the top point of the high-frequency arc can be achieved.

By considering the measured sample dimensions, the bulk resistivity can be calculated. In Figure 18a, the plot values of bulk resistivity versus the samples’ Ksil content are reported. In Figure 18b, the relaxation frequency is plotted versus Ksil content. The potassium silicate activator (Ksil) concentration in Figure 18 is reported in terms of the weight% with respect to the initial weight, which is the weight of all the reagents, including any added extra water.

In Figure 18a, an increase of resistivity with increasing Ksil content is shown: this behavior can be explained by the fact that for geopolymer formulations, in general, exists an optimum concentration of the activator to reach the best mechanical performances, which occur when the concentration of alkali activator ions (K^+^ in our case) equals the concentration of Al^3+^ ions of the aluminum–silicate network of the forming geopolymer [16]. The presence of K^+^ excess leaves unreacted activator that eventually carbonates and can lead to a decrease in mechanical properties due to incomplete geopolymerization. However, due to the particular nature of the samples, which were prepared with the lowest possible fluidity to avoid magnetic powder aggregation, different levels of geopolymerization (depending on activator and water amounts) can be achieved. Since electrical resistivity of geopolymers is known to increase with geopolymerization [74], incomplete geopolymerization is expected to yield a lower electrical resistivity as is observed in G1is_6_norm, which contains a similar concentration of Ksil as the matrix, but under low fluidity under adverse mixing conditions. In sample G1is_6_highK, the higher concentration of Ksil contributes to better geopolymerization both directly via a higher dissolution rate of the MK and slag reagents and indirectly due to a slightly better fluidity coming from the liquid part of the Ksil, thus compensating for the adverse mixing conditions, resulting in a sample with electrical resistivity slightly higher than the matrix. For the sample G1is_6_highW in which fluidity and mixing conditions were improved via extra water addition, reactivity and geopolymerization should be similar to the matrix, even if the extra water addition increases sample porosity, which is known to decrease electrical resistivity [75,76], in agreement with the lower resistivity observed for sample G1is_6_highW.

Figure 18b shows the so-called “relaxation frequency” as a function of Ksil concentration. Relaxation frequency is related to the double-layer capacitance of the sample associated with the solid/pore solution interfaces in the sample. In analogy to concrete and mortar samples, a lower relaxation frequency indicates a higher level of polymerization [75]. Hence the observed trend of relaxation frequency confirms an increased geopolymerization with increasing Ksil concentration in our samples.

Concerning the available literature, conductivity values for geopolymers span a wide range, depending on the reagents used and preparation conditions. For example, conductivity values reported in. [76] for Nasil-blast furnace slag samples aged 30 days are 0.02–0.05 S/m, while in. [77] for Mk-Nasil geopolymer samples, a value of 6.2·10^−3^ S/m can be retrieved based on reported sample dimensions and resistance values. In ref. [78], for geopolymer materials prepared using sodium silicate solutions and chemosynthetic Al_2_O_3_–2SiO_2_ amorphous powders, a value of 1.5·10^−4^ S/m is reported. In comparison, our samples exhibit conductivities ranging from 4.4–8.3·10^−4^ S/m, which is about 10–100 times lower than values reported in refs. [76,77] and 3–6 times higher than that of chemosynthetic geopolymers in [78]. Our highest value of resistivity (2.3·10^5^ ohm-cm, for G1is_6_highK) is in the same range as given by Davidovits, [13] suggesting a complete geopolymerization for this sample.

### 3.4. Mechanical Properties

The compressive strength data (Table 9) at 7 days of samples without quartz aggregates are reported in Figure 19, and a decrease with the increasing addition of magnetic powder can be noticed.

Since the extra water addition was kept quite constant at 10% for samples G1, G1_6, and G1_11 (both “is” and “anis”), the strength decrease must be associated directly with the magnetic particle addition. For samples G1_6DENSE, which were prepared with lower extra water than G1_6, higher compressive strength is observed, comparable to that of the matrix. Thus, in the DENSE samples, the negative effect of the presence of magnetic particles seems to be counterbalanced by the reduced amount of extra water used (2.3–3.3%).

For the sample set G2, where quartz aggregates are added to the matrix to reduce the risk of aggregation of the magnetic particles, the compressive strength results shown in Figure 20 evidence an initial increase in strength as the magnetic powder was added to the quartz aggregates. As more magnetic particles are added, the strength decreases. This behavior can be explained by considering both as result of the presence of the magnetic particles and of the different amounts of extra water added during preparation. It is to note a slight difference between the values obtained when the isotropic particles are present, which are higher, concerning the anisotropic ones, reasonably due to a less inhomogeneous distribution of the magnetic particles inside the matrix, where larger anisotropic particles aggregates behave as places of fragility.

Comparing G1 and G2 series samples, two observations can be made: (1) A higher strength of matrix G1 is observed compared to matrix G2. This is reasonably due to the higher fluidity and thus better mixing conditions for matrix G1 due to the absence of aggregates, which promotes a higher degree of geopolymerization, as observed in the electrical measurements. (2) The strength difference between samples with isotropic and anisotropic magnetic powders is lower for G1 series samples compared to G2 series samples. As already noted, this strength difference is related to the different tendencies of clustering of the isotropic and anisotropic magnetic particles used. Since the addition of fine-grained quartz aggregates in G2 series samples was intended to reduce such magnetic particle clustering, the result appears surprising at first. However, it needs to be considered that the addition of fine quartz aggregates also has the side-effect of increasing the available surface area of aggregates to be infiltrated by the matrix. Therefore, less matrix is available to bind the additional magnetic particles, and any residual magnetic particle clusters will be weaker bonded in samples with quartz aggregates than in samples without. Hence, considering the different clustering tendencies of the isotropic and anisotropic particles used, the strength difference could well be amplified in samples with quartz aggregates if the dispersion were not effective enough.

It is noteworthy that the values of compressive strength here observed at 7 days resulted to be higher concerning those reported for high weight concretes bearing high content of magnetite (< 30 MPa) [7,14] and for NdFeB/cement composites [79].

## 4. Conclusions

A series of novel magnetic geopolymers were prepared, and many chemical, morphological and physical characterization data are collected to investigate the effect of the presence of SrFe_12_O_19_ particles.

The experimental procedure for preparation appears to be crucial to determine the morphology and the electrical properties of the final composites, in particular in terms of extra water amount and mixing. Thus, in principle, the preparation can be optimized even in the presence of higher amounts of magnetic particles, in terms of a suitable ratio between reactive components even in the presence of higher amounts of water necessary for good mixing.

The magnetic study has revealed that the SrFe_12_O_19_ magnetic particles can be relatively well dispersed in the inorganic matrix. A substantial increase in the remanent magnetization of the composite samples above the value achievable in the case of a random spatial distribution of the particles was obtained by embedding in the geopolymer SrFe_12_O_19_ anisotropic particles at high concentration, under the action of an external magnetic field during the solidification process.

The impedance measurements indicated that electrical resistance is mainly controlled by the chemical composition of the matrix as well as mixing conditions, yielding a maximum resistivity value (as shown by the diameters of the high-frequency arc in Nyquist plots) of 2.3·10^5^ ohm cm for the sample G1is_6_highK, which is in the same range as given by Davidovits, [13] indicating for the sample a complete geopolymerization.

The collected data have demonstrated that new composites exhibit good mechanical properties (as compressive strength), higher with respect to those reported for high weight concretes with a high content of magnetite.

The work is ongoing with the objective to better control the final configuration of the magnetic particles into the inorganic matrix and to promote the participation of magnetic particles, suitably functionalized, to the buildup of the geopolymer matrix, to prepare materials with designed mechanical, magnetic and electrical properties.

The method could be of general application, considering the wide range of experimental parameters and modulability of the geopolymers preparation, together with the different types and amounts of magnetic particles, which could be introduced into the inorganic matrix, together with the fact that by using geopolymerization strategy exotic shapes to the final products could be achieved for a wide range of applications.

## Data Availability

All the available data have been reported either in the manuscript or in the supporting materials.

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
