# Peer review of "Novel Magnetic Inorganic Composites: Synthesis and Characterization"

_polymers, 2021, doi:10.3390/polym13081284_

Round 1

Reviewer 1 Report

The paper describes the fabrication and characterization of magnetic geopolymer composites. The authors compare the nanocomposites obtained using isotropic and anisotropic magnetic fillers, fabricated with or without external magnetic field. The results show clearly the influence of these conditions on the dispersion state and magnetic properties of the composites.
I found the paper interesting and well-written and would recommend the publication if the following comments are properly addressed:
1) In the introduction, the authors should explain the main aim of this study and the structure of the paper; this would help to make the links between different sections;
2) I would appreciate if the authors cite more references on the reinforcement of geopolymers by fillers showing the influence of the filler dispersion state on the mechanical properties;
3) Fig. 6 and 7 add the full legend explaining the meaning of the symbols on the spectra; modify the axis ticks in a way to facilitate the location of the peaks;
4) line 439 - "particles aggregate ... in a lower extent": could the authors make more quantitative analysis?
5) line 455 - the sentence "No differences are found...": could the authors make a comment about what it means regarding the polycrystalline nature of the particles?
6) Fig 14 - mark all the peaks including the matrix; adapt the ticks on the x axis to help locating the peaks;
7) in general, the axis limits of XRD patterns are arbitrary; I would recommend to homogenize and to give the full-scale XRD patterns;
8) line 489 - the sentence "Moreover, Hc is higher": please, give a reference or explain more in detail the effect of interparticle interactions;
9) line 513 - the paragraph is discussing the sample G1anisM_11: could the authors provide an ESEM image for this sample?
10) the same paragraph - why the Hc is lower in the sample with G1 matrix with respect to G1 one? does the quartz filler block the particle rotation?
11) line 522 - the sentence "Anisotropic particles, larger ...": could the authors add a reference about the magnetic torque force acting on anisotropic particles?
12) section 3.3.4. - in the beginning, explain the aim of measuring of nanocomposite conductivity;
13) figure 17 - define the "fresh weight";
Minor tips:
line 37 - add "magnetite (Fe3O4)";
line 49 - I don't see the need of the paragraph citing the ref 9;
line 157 - "Scanning" instead of "Scansion" and remove the 2nd "embedded";
line 257 - cite the Fig 1c and d;
page13 - use the term XRD "patterns" instead of "spectra";
Fig 3 - Cobalt or copper anode? (homogenize with the methods section);
lines 615 and 619 - "with respect to"
line 545 - remind what is Ksil;

Author Response

Response to Reviewer 1 comments

The paper describes the fabrication and characterization of magnetic geopolymer composites. The authors compare the nanocomposites obtained using isotropic and anisotropic magnetic fillers, fabricated with or without external magnetic field. The results show clearly the influence of these conditions on the dispersion state and magnetic properties of the composites.

I found the paper interesting and well-written and would recommend the publication if the following comments are properly addressed:

Reply:

We thank the reviewer for the appreciation of our work and the time and expertise dedicated to the revision of our manuscript. We reviewed the text according to her/his useful suggestion.

  • In the introduction, the authors should explain the main aim of this study and the structure of the paper; this would help to make the links between different sections;

Reply:

We thank the Reviewer for her/his suggestion. We provided a brief description of the aim of our work in the Introduction by modifying the last paragraph as follow: “In the present work, the preparation of novel magnetic geopolymers, based on two different matrices containing commercial SrFe12O19 particles, with two different concentrations, has been described. The objective is the investigation of the influence of composition and experimental conditions for preparation on the morphology, dispersion of SrFe12O19 particles, electrical and mechanical properties. Particularly attractive is the study of the magnetic properties of the composites and of their modification when solidification of the geopolymers is performed in the presence of an external magnetic field. The target of the research project is to design magnetic inorganic composites to apply for pollutant removal [32,33,34] and heterogeneous catalytic processes [35] on large scale.”

  • I would appreciate if the authors cite more references on the reinforcement of geopolymers by fillers showing the influence of the filler dispersion state on the mechanical properties;

Reply:

As highlighted by the Reviewer, the Introduction was lacking in providing a background to the reinforcement of geopolymers by fillers and their influence on the composite mechanical properties. We added the following paragraph to the Introduction: “As for the mechanical properties, it is known that they are improved by the incorporation and the homogeneous distribution of nanomaterials due to the formation of a denser microstructure, but the identification of the optimum content of nanomaterials is very important to achieve the respective high performance in strength as well as workability [20]. In the case of the addition of nanosized Fe2O3, it was reported that in Portland cement composites a homogeneous microstructure was obtained with magnetite loading lower than 10% w/w with an improvement of the compressive strength by 60%. Higher added amounts produced aggregation of the nanoparticles with a detrimental effect on the mechanical performance of the material [9].” with a new reference [20].

  • 6 and 7 add the full legend explaining the meaning of the symbols on the spectra; modify the axis ticks in a way to facilitate the location of the peaks;

Reply:

The caption of Figure 6 and 7 have been revised and the symbols explained. We also modified the axis ticks to make the location of the peaks easier for the reader.

  • line 439 - "particles aggregate ... in a lower extent": could the authors make more quantitative analysis?

Reply:

Unfortunately, we did not perform an extensive quantitative analysis of the particle aggregation in the composites, nevertheless we added semi-quantitative size information on the aggregates based on the ESEM images collected. The text after Figure 13 was modified as follows: “Magnetic particles aggregate also in the G2isM_6 sample (Figures 13c), but in a lower extent, giving rise to smaller aggregates (typically 20 x 20 μm for is-SrFe12O19 vs 30x30 μm for anis-SrFe12O19, in the presence of the same amount of ferrite, in more distributed shape) and appear to be more distributed in the case of G2is_6, where rare, ordered aggregates can be detected (Figures 13d).”

  • line 455 - the sentence "No differences are found...": could the authors make a comment about what it means regarding the polycrystalline nature of the particles?

Reply:

We agree with the Reviewer that the sentence: “No differences are found either in the samples solidified without or under magnetic field, or in the case of composites containing isotropic or anisotropic ferrites.” was too generic, roughly qualitative, and somewhat clashing with the observations reported in the text, so we preferred to remove it. The similar sentence: “It is to note that XRD profiles do not evidence difference either in the samples solidified without or under magnetic field, or in the case of composites containing isotropic or anisotropic ferrites.” has also been removed from the Conclusions.

  • Fig 14 - mark all the peaks including the matrix; adapt the ticks on the x axis to help locating the peaks;

Reply:

All the relevant peaks, including the matrix, have been marked and the ticks on the x axes adapted according to the Reviewer suggestion.

  • in general, the axis limits of XRD patterns are arbitrary; I would recommend to homogenize and to give the full-scale XRD patterns;

Reply:

The axes limits have been homogenized. To make the more relevant information clearer in the XRD patterns reported in the manuscript, we made the full-scale ones available in the Supporting Materials.

  • line 489 - the sentence "Moreover, Hc is higher": please, give a reference or explain more in detail the effect of interparticle interactions;

Reply:

We thank the Reviewer for her/his request, which prompted us to discuss in more detail the influence of interparticle magnetic interactions on the magnetic hysteretic properties of the investigated samples, in particular on the coercivity. The matter is complex because both exchange and dipolar interactions are likely to exist, but their specific role in determining the overall magnetic behaviour of the samples is substantially impossible to disentangle.

The sentence indicated by the Reviewer refers to the magnetic geopolymer samples, but, to clarify the meaning, we have better dealt with the magnetic properties of the starting isotropic and anisotropic Sr-ferrite particles, including the particular shape of the hysteresis loops (wasp-waist-like). Therefore, the part of Section 3.1 about the magnetic characterization of the starting particles has been considerably expanded (pages 9 – 11 of the revised version of the manuscript) and 9 new References have been added.

Then, in Section 3.3.3, page 20 and 21 of the revised version of the manuscript, we have better commented on the shape of the hysteresis loops in the composite samples, which is different from that of the starting particles. The sentence indicated by the Reviewer, now reported on page 21, has not been changed, actually. In fact, we hope and are quite confident that, in light of the explanations added in the manuscript, particularly the one at the end of Section 3.1, its meaning is now clear.

  • line 513 - the paragraph is discussing the sample G1anisM_11: could the authors provide an ESEM image for this sample?

Reply:

We provided the ESEM image of composite G1anisM_11 as the new Figure 16, in which the dispersion of the ferrite particle is clearly visible.

  • the same paragraph - why the Hc is lower in the sample with G1 matrix with respect to G1 one? does the quartz filler block the particle rotation?

Reply:

In her/his question, the Reviewer has possibly written “… G1 matrix with respect to G1 one” in place of “… G2 matrix with respect to G1 one”. So, we assume that she/he is making the comparison between the ‘anis’ geopolymer samples, G2anis_M6 and G1anis_M11, due to the large difference between their Hc values.

The difference between the samples is not just related to the matrix, but also to the particles concentration, as the sample with the highest Hc value, G1anis_M11, has a 11 % anisotropic particles concentration, whilst G2anis_M6 has a 6 % one. Due to that, the Lorentz field contribution (as explained in the manuscript, lines 520 – 524 of the original manuscript), is smaller in G2anis_M6 with respect to G1anis_M11, so in G1anis_M11 the particles reorientation process appears to be more effective.

Moreover, apart from G1anis_M11, all the samples of the ‘M’ group display a similar Hc value, irrespective of the matrix and of the particles concentration. Hence, the smaller Hc value of G2anis_M6, with respect to G1anis_M11, should not be due to the hindering of particles rotation induced by the quartz filler during the curing process.

  • line 522 - the sentence "Anisotropic particles, larger ...": could the authors add a reference about the magnetic torque force acting on anisotropic particles?

Reply:

In a magnetic particle with a high magnetic anisotropy (shape or magnetocrystalline in nature), which is our case, a strong coupling is established between the magnetic moment and the body of the particle. Therefore, an applied magnetic field at first magnetizes the particle to a certain extent and then exerts a torque on the net magnetic moment acquired by the particle. Thus, a physical rotation of the particle can be observed, provided that it is free to move. In this regard, a new reference has been included in the manuscript (Erb, R.M.; Martin, J.J.; Soheilian, R.; Pan, C.; Barber, J.R. Actuating Soft Matter with Magnetic Torque. Adv. Funct. Mater. 2016, 26, 3859–3880).

As stated in the sentence indicated by the Reviewer, we expect that, under the same magnetic field HAPP, the magnetic moment acquired by the anisotropic particles is greater, because they are on average larger and have higher MS compared to the isotropic ones. Therefore, the intensity of the torque, which is obviously proportional to the particle magnetic moment, is higher for the anisotropic particles, which implies that HAPP exerts a more effective rotation action on the anisotropic particles.

  • section 3.3.4. - in the beginning, explain the aim of measuring of nanocomposite conductivity;

Reply:

The aim of measuring the conductivity of the produced composites comes from the desire to shed light on the interaction of the composite geopolymer with electromagnetic fields, for a more precise characterization of these materials, and to account for possible applications in which they undergo the effect of electrical fields. We added an introductory line to the section 3.3.4.

  • figure 17 - define the "fresh weight";

Reply:

In the caption of figure 17 and in the text we changed “fresh” with “initial” and explained the meaning of the expression as follows: “The potassium silicate activator (Ksil) concentration in fig 17a,b is reported in terms of the weight% with respect to the initial weight, which is the weight of the all reagents, including any added extra water.”

Minor tips:

line 37 - add "magnetite (Fe3O4)";

line 49 - I don't see the need of the paragraph citing the ref 9;

line 157 - "Scanning" instead of "Scansion" and remove the 2nd "embedded";

line 257 - cite the Fig 1c and d;

page13 - use the term XRD "patterns" instead of "spectra";

Fig 3 - Cobalt or copper anode? (homogenize with the methods section);

lines 615 and 619 - "with respect to"

line 545 - remind what is Ksil;

Reply:

 All the minor tips listed by the Reviewer have been taken into account, corrections made, redundant paragraph removed, and clarifications given in the text. Every modification has been highlighted in yellow in the revised version of the manuscript.

Reviewer 2 Report

This manuscript is well-organized, and the contents are in-details. I suggest its publication after minor revisions. My comments are as below:

  1. The keywords should be re-written to make them more standardized.
  2. In the second paragraph of introduction section, the authors state that “Concretes bearing different amounts of Fe3O4 or Fe3O4@SiO2 nanoparticles behave 40 also as electromagnetic wave absorbers (particularly of microwaves) and show γ-rays 41 shielding properties, thus featuring a huge potential application in construction of elec-42 tromagnetic wave interference buildings.” This statement is good. However, the supporting references are poor and not enough. More references with high impact should be cited to displace them. For example, J Alloy Compd 2021 858: 157706; Chem Eng J 2020 391: 123571; Compos Part B-Eng 2021 214: 108744; et al.
  3. The mechanical properties should be compared between G1 and G2, and the corresponding mechanism should be given in details.
  4. The differences of the Nyquist plots of the samples as shown in Figure 16 should be discussed more detailed.
  5. What’s the difference between the data in Table 7 and Table 8?
  6. Figure 12 and Figure 13 can be put together.
  7. All the samples should be put into a table to show what is the meaning of the subscript of their names.

Author Response

Response to Reviewer 2 comments

This manuscript is well-organized, and the contents are in-details. I suggest its publication after minor revisions.

Reply:

We thank the reviewer for the appreciation of our work and the time and expertise dedicated to the revision of our manuscript. We reviewed the text according to her/his useful suggestion.

My comments are as below:

  • The keywords should be re-written to make them more standardized.

Reply:

Thanks to the Reviewer suggestion, we revised the keywords as follow: “Magnetic geopolymer composites; SrFe12O19 particles; Morphological characterization; Magnetic properties; Impedance measurements.”

  • In the second paragraph of introduction section (lines 40-42), the authors state that “Concretes bearing different amounts of Fe3O4 or Fe3O4@SiO2 nanoparticles behave also as electromagnetic wave absorbers (particularly of microwaves) and show γ-rays shielding properties, thus featuring a huge potential application in construction of electromagnetic wave interference buildings.” This statement is good. However, the supporting references are poor and not enough. More references with high impact should be cited to displace them. For example, J Alloy Compd 2021 858: 157706; Chem Eng J 2020 391: 123571; Compos Part B-Eng 2021 214: 108744; et al.

Reply:

We agree with the Reviewer’s observation. We added the following paragraph in the Introduction “An intriguing advantage derives from the synthetic versatility of concretes and geo-polymers in terms of the possible introduction of a wide variety of additives able to absorb or dissipate electromagnetic waves [10,11,12].” with the suggested references.

  • The mechanical properties should be compared between G1 and G2, and the corresponding mechanism should be given in details.

Reply:

We thank the Reviewer for the comment which allowed us to improve the description of the mechanical properties of the composites we present. The following section has been added to section 3.6: “Comparing G1 and G2 series samples two observations can be made: 1) A higher strength of matrix G1 is observed compared to matrix G2. This is reasonably due to the higher fluidity and thus better mixing conditions for matrix G1, due to the absence of aggregates, which promotes a higher degree of geopolymerization, as observed in the electrical measurements. 2) The strength difference between samples with isotropic and anisotropic magnetic powders is lower for G1 series samples compared to G2 series samples. As already noted, this strength difference is related to the different tendency of clustering of the isotropic and anisotropic magnetic particles used. Since the addition of fine-grained quartz aggregates in G2 series samples was intended to reduce such magnetic particle clustering the result appears surprising at first. However, it needs to be considered that the addition of fine quartz aggregates has also the side-effect of increasing the available surface area of aggregates to be infiltrated by the matrix. Therefore, less matrix is available to bind the additional magnetic particles and any residual magnetic particle clusters will be more weakly bonded in samples with quartz aggregates than in samples without. Hence, considering the different clustering tendency of the isotropic and anisotropic particles used, the strength difference could well be amplified in samples with quartz aggregates if the dispersion were not effective enough.”

  • The differences of the Nyquist plots of the samples as shown in Figure 16 should be discussed more detailed.

Reply:

We thank the Reviewer for observing that these figures were not analysed completely. We thus commented on the additional information contained in the Nyquist plots that is the “depression angle”. While in principle this parameter is potentially useful to analyse trends in materials properties its small variation observed in our samples does not allow to discuss any changes in material properties in our case.

The following sentence has been added to the manuscript: “In the plots, the high frequency arcs are well outlined. In all cases the arc centre is depressed below the real axis, which is due to relaxation of polarization processes within the material and results in dielectric dispersion, i.e., decrease of the capacitance when increasing the frequency [73]. The depression angle is in the range 23° - 27° for the four plots reported.”

  • What’s the difference between the data in Table 7 and Table 8?

Reply

Table 8 includes Hc (coercivity) and MR (squareness) data for some of the geopolymer samples that have been prepared by applying an external magnetic field, Happ, during the curing process. In fact, this was indicated in the text of the original manuscript (lines 495-497), close to Table 8, but was not included in the caption of Table 8. To better clarify this point, the captions of Table 7 and Table 8 have been modified as follows:

  • Table 7. The previous caption “Magnetic data relative to the samples labelled as in Column 1.” has been modified into “Magnetic data relative to the composite geopolymer samples, labelled as in Column 1.”
  • Table 8. The previous caption “Magnetic data relative to the samples labelled as in Column 1 at 300 K.” has been modified into “Magnetic data relative to the composite geopolymer samples, labelled as in Column 1, prepared in the presence of a magnetic field Happ during the curing process.”

  • Figure 12 and Figure 13 can be put together.

Reply

We thank the reviewer for the suggestion, but we decided to split the two figures to avoid having a too big one that would have forced us to reduce the size of all the ESEM images. For these reasons we still prefer to keep them separate. If the Reviewer or the Editor still wish to merge them into one, we will follow their recommendation.

  • All the samples should be put into a table to show what is the meaning of the subscript of their names.

Reply

We thank the reviewer for the suggestion. We added a naming scheme as Scheme 1 to make clearer the meaning of the nomenclature used throughout the text.
